# Genome-Wide Identification and Expression Profiling of Glycosidases, Lipases, and Proteases from Invasive Asian Palm Weevil, *Rhynchophorus ferrugineus*

**DOI:** 10.3390/insects16040421

**Published:** 2025-04-17

**Authors:** Nazmi Harith-Fadzilah, Mohammad Nihad, Mohammed Ali AlSaleh, Abdulqader Yaslam Bazeyad, Subash-Babu Pandurangan, Kashif Munawar, Arya Vidyawan, Hattan A. Alharbi, Jernej Jakše, Arnab Pain, Binu Antony

**Affiliations:** 1School of Agriculture Sciences and Biotechnology, Faculty of Bioresources and Food Industry, Universiti Sultan Zainal Abidin, Besut 22200, Malaysia; 2Department of Plant Protection, Center for Chemical Ecology and Functional Genomics, College of Food and Agricultural Sciences, King Saud University, Riyadh 11451, Saudi Arabia; 3Department of Food Science and Nutrition, College of Food and Agricultural Sciences, King Saud University, Riyadh 11451, Saudi Arabia; 4Agronomy Department, Biotechnical Faculty, University of Ljubljana, SI-1000 Ljubljana, Slovenia; 5Pathogen Genomics Group, Bioscience Program, Biological and Environmental Science and Engineering (BESE) Division, King Abdullah University of Science and Technology (KAUST), Thuwal, Jeddah 23955, Saudi Arabia; arnab.pain@kaust.edu.sa

**Keywords:** red palm weevil, herbivory, digestive enzyme, gene duplication, phylogeny, molecular docking

## Abstract

The red palm weevil is an invasive weevil that threatens many types of palm trees worldwide. Conventional chemical insecticides used to control it can harm other creatures, humans, and the environment. Currently, researchers are seeking small molecules that can selectively suppress the activity of proteins essential for the weevil’s survival. One promising target is the weevil’s digestive enzymes, which are vital for feeding. To gather an understanding of the digestive enzymes in *R. ferrugineus*, we used a bioinformatic approach to identify all the glucosidases, lipases, and protease genes from its genome and transcriptome data. We identified gene transcripts of the red palm weevil’s gut and abdomen, discovering 34 carbohydrate-, 85 lipid-, and 201 protein-digesting enzymes. Genome-wide analysis revealed that several key digestive enzymes have predominantly emerged through gene duplication. Furthermore, the modeling and molecular docking of select enzymes revealed that these enzymes had high structural similarity with existing enzymes in the Protein Data Bank and demonstrated similar ligand-binding profiles compared to their corresponding reference template enzymes. The knowledge in this study provides a basis for developing new, eco-friendly, and selective red palm weevil insecticides, which offer a safer and more precise pest control method.

## 1. Introduction

Over four million species of beetles (Order Coleoptera) are known at the moment, constituting the most species-rich animal group on earth. Their diversity is mainly attributed to their plant-feeding (herbivory) habit, as a result of an extensive diversification of plant cell wall-degrading enzymes, alongside the functional specialization of key digestive enzymes [1,2]. Digestive enzymes (DEs) are vital for facilitating feeding and nutrient acquisition. DEs are produced primarily by the salivary gland and the gut and are broadly categorized into glycosidases, proteinases, and lipases [3]. Glycosidases are vital for cell wall and carbohydrate digestion, allowing access to plant cells while also deriving nutrients from the cell wall. Proteinases break down proteins during digestion, while some groups of proteinases have additional vital roles in insect growth and in preserving nervous system function [4]. Lipases are involved in lipid digestion and have additional roles in ensuring proper insect development, neurotransmission, and insect immunity [5].

The *Rhynchophori* weevils (palm weevils) are herbivorous insect pests that attack palm trees globally. These weevils originated from South Asia; their high mobility and adaptability allow them to infest other palm trees not native to the region and quickly become a severe threat to the affected places [6,7]. Among the *Rhynchophori* weevils, the Asian palm weevil (also known as the red palm weevil), *Rhynchophorus ferrugineus*, is the most invasive, quarantine category-1 pest [8], has been known to infest economically significant palms such as the coconut, date palm, Canary Island date palms, oil palms, sago palms, and a wide range of ornamental palm species [9]. The specimens in the larval stage inflict the most significant damage by chewing the inside of the host palm tree’s structure, destroying the vascular system [7,10]. This weakened structure eventually causes the mortality of the host tree. *R. ferrugineus* outbreaks are managed via effective monitoring and early detection approaches, coupled with the administration of hazardous chemical insecticides potentially harmful to non-target insects, humans, and the surrounding environment [11].

As an alternative, more recent research efforts have focused on developing weevil management strategies targeting insect digestive enzymes, providing new options for pest management in the era of synthetic biology and biotechnology. This research utilizes contemporary biological knowledge of insect digestion based on the proteins corresponding to critical life processes such as herbivory, growth, and mating. These proteins may subsequently be targeted for disruption through chemical perturbation, gene silencing, and gene editing approaches to induce mortality or disrupt digestive processes. Furthermore, contemporary insect control research explores using natural secondary metabolites derived from plants, microbes, and other organisms to derive and develop more toxicologically and environmentally benign insecticides. For example, the use of α-amylase inhibitors derived from plant sources was explored for suppressing the amylolytic activity of insects [12,13]. Alternatively, crops could be genetically engineered to produce small-molecule inhibitors, potentially providing resistance against weevils [14]. The most well-known example is transgenic Bt corn, which contains *Bacillus thuringiensis* Bt Cry toxins, which induce insect mortality by inhibiting aminopeptidase N (APN) activity [15]. Furthermore, toosendanin, a bioactive ingredient from a commercialized insecticide, disrupts lipase activity within the insect midgut with the further consequence of disrupting neurotransmission and metamorphosis-related processes (https://pubchem.ncbi.nlm.nih.gov/compound/Toosendanin, date accessed: 18 March 2025).

To acquire promising DE leads for in vitro and in vivo characterization analyses from tens of thousands of insect proteins, an omics approach coupled with genome or proteome annotation in silico can narrow down the genes or proteins of interest in high throughput. The transcriptomics method produces and analyses expressed genes from a biological sample in a high-throughput fashion. With structural information, DEs can become a promising target to inhibit activity, disrupting herbivory and consequently inducing mortality. Such research efforts can then be translated into new insecticide formulations or the generation of transgenic crops that are specifically resistant to the insect pest of interest.

This study aimed to identify, annotate, and profile the gene expression of *R. ferrugineus* DEs and perform genome-wide analyses using field sampling and laboratory rearing through the transcriptomics approach. The selected DEs were examined in bioinformatic analyses to assess their similarity with other related coleopterans and evaluate the ligand-binding performance of DEs of interest via molecular docking. These DEs of interest may serve as a potential new target for selective inhibition against *R. ferrugineus*.

## 2. Materials and Methods

### 2.1. Insect Collection and Tissue Dissection

Red palm weevil adults were collected in 2009 with the direct permission of a cooperating landowner [Al-Kharj region (24.1500° N, 47.3000° E) of Saudi Arabia]. Since then, the laboratory colony has been maintained, as previously described [16,17,18]. The weevils were maintained on sugarcane stem diets under rearing conditions at a 23 ± 2 °C temperature, 30 ± 4% RH, and 14:10 h (D:L) photoperiod [16,17,18]. The RPW male and female adults from the field were captured alive in May 2021 from infested and removed date palm tree materials in Al-Kharj in Saudi Arabia, designated as field-collected samples. Insects were anesthetized using CO_2_ for 1–2 min, and the abdomen and gut were carefully dissected (males and females separately) under a light microscope. Immediately after collection, the tissues were transferred to an RNALater solution and stored at −20 °C until total RNA extraction. Approximately 30 mg tissues were collected from each group, transferred in an RNA storage solution, and then stored at −80 °C until total RNA extraction.

### 2.2. Total RNA Extraction, cDNA Library Construction, and Sequencing

Following the manufacturer’s instructions, a single replicate from the larva and adult gut group was used to extract the total RNA using the PureLink RNA Mini Kit (Thermo Fisher, Bedford, MA, USA). A NanoDrop spectrophotometer (Thermo Fisher, MA, USA) was used to quantify and check the quality of the extracted RNA and synthesized cDNA. The quantity and quality of the total RNA were validated using a Qubit 2.0 Fluorometer (Thermo Fisher, Bedford, MA, USA), and RNA integrity was confirmed using a 2100 Bioanalyzer (Agilent Technologies, Santa Clara, CA, USA). After the quality and the characteristic “hidden break” in the 28S RNA profile were ensured using the 2100 Bioanalyzer [19], a paired-end cDNA library was prepared using the TruSeq Stranded mRNA library preparation Kit (Illumina, San Diego, CA, USA) following the manufacturer’s protocols, which include the following steps: purification and fragmentation of total RNA, first- and second-strand cDNA synthesis, 3′-end adenylation, adapter ligation, and purification. Finally, the purified and PCR-enriched products were used for cDNA library preparation. The cDNA libraries were validated and quantified by the Qubit 2.0 Fluorometer. Illumina HiSeq (model: 2000) sequencing was performed at the core sequencing facility of the King Abdullah University of Science and Technology (KAUST), Jeddah, Saudi Arabia.

### 2.3. Data Processing, Assembly and Annotation, and Differential Expression Analysis

Image deconvolution and quality value calculations were performed using Illumina GAPipeline1.3. Illumina adaptors were detected and removed by an automatic read-through adapter trimming option implemented in the “Trim Reads” tool of the Qiagen CLC Genomics Server (CLC) (v. 21.0.1). Low-quality bases were also trimmed off, allowing two ambiguities per read. Filtered paired-end reads were QC-validated through the “QC for Sequencing Reads Tool” of CLC. A reference de novo transcriptome assembly was constructed with the “De Novo Assembly Tool” of CLC and contigs functionally annotated by the BLAST2GO command line tool (v. 1.5). DE transcript-level quantification was performed to identify the differentially expressed genes. The transcripts per kilobase per million mapped reads (TPM) value of each DE gene was calculated. The normalized transformed TPM values were tabulated and converted to heatmaps using R and R Studio (v. 2023.03.0+386 “Cherry Blossom”).

### 2.4. Phylogenetic and Genome-Wide Analysis of the Candidate Digestive Enzymes

A phylogenetic tree was made separately for glycosidases, lipases, and proteases. For proteases, only aspartic proteases were selected for the construction of the phylogenetic tree due to the large number of proteases identified. Other closely related beetles and weevils, namely, *R. ferrugineus* (Rfer), *Anoplophora glabripennis* (Agla), *Dendroctonus ponderosae* (Dpon), *Leptinotarsa decemlioneata* (Ldec), and *Sitophilus oryzae* (Sory), were used for comparison, with an additional *B. mori* (Bmor) as the outgroup. All the predicted sequences were aligned using MAFFt v. 7 [20], utilizing the default method with all parameters at default settings, followed by manual trimming to remove gaps and ambiguity bases. The auto algorithm and BLOSUM62 were used as the scoring matrix. An automatic model search was performed using ModelFInder [21], of which the software identified LG+F+I+G4, WAG+G4, and LG+I+G4 substitution models were used for aminopeptidase, amylase, and lipase, respectively, according to the Bayesian Information Criterion (BIC) best-fit model. A maximum likelihood analysis was performed using default settings and ultrafast bootstrap support with 1000 replicates using IQ-tree [22]. The tree was rooted, visualized, and edited with FigTree v1.4 (tree.bio.ed.ac.uk), colored, and finally edited with Adobe Illustrator v. 29.3 (Adobe, San Jose, CA, USA).

The DE sequences were manually annotated and mapped to the *R. ferrugineus* genome [23] (GenBank accession numbers GCA_014462685.1 and GCA_014490705.1) using a BLASTn search against the *R. ferrugineus* genome, which was created using Geneious v7.1.9 (Biomatters, Auckland, New Zealand). The exon-intron positions of the DEs in the genome were mapped at the scaffold region in a different locus in the *R. ferrugineus* genome. The mapped regions were extracted and manually aligned using the MAFFT program v. 7.38, which was used for gene structure illustrations. The NCBI graphical sequence viewer (v. 3.50.0), available in the NCBI Genome Workbench, was used to graphically display the nucleotide and protein sequences at the scaffold region in a different locus. To generate assemblies of RferLip and RferPro alleles, we mapped gene sequences onto the *R. ferrugineus* genome and identified locus tags and scaffolds. We then verified that the allelic sequences matched through manual sequence alignments using the MAFFT program. Each allele alignment was classified into one of three classes based on the variation between alleles: (1) single-site substitutions, (2) small indels, and (3) complete repeat indels (CRIs), defined as those indels that encompassed one or more complete repeat units. The amino acid sequence similarity of DE genes was tested with the Psi-BLAST (NCBI) sequence alignment algorithm based on the e-value, bit-score, and percent identity. An attempt to infer duplication events through unrooted protein trees for the DEs was made using BLAST pairwise alignment in the NCBI Tree Viewer (v. 7.1.0.46). The DE protein sequence similarity search using the BLASTp e-value with a cutoff expectation of <2 and <10^3^ identified different DE clades, shedding light on gene duplication events.

### 2.5. Structural Modeling and Molecular Docking

The enzyme with the most abundant mRNA transcript from each group of DEs selected for phylogenetic tree reconstruction was modeled using AlphaFold3 [24]. The FASTA sequences of the selected glycosidases, lipases, and proteases were submitted to the AlphaFold3 web server (https://alphafoldserver.com/welcome, accessed on 10 June 2024). Modeled proteins were evaluated via MolProbity (http://molprobity.biochem.duke.edu/, accessed on 10 June 2024) to evaluate the MolProbity score, via SAVES (https://saves.mbi.ucla.edu/, accessed on 10 June 2024) for ERRAT, Verify3D, and Procheck sores, via Swiss-Model (http://molprobity.biochem.duke.edu/, accessed on 10 June 2024) for QMeanDisco Global scores [25,26,27,28,29], and the Simple Modular Architecture Research Tool (SMART) (http://smart.embl-heidelberg.de/, accessed on 10 June 2024) to validate each modeled DE’s protein identity based on the identified domains from the SMART scan [30].

We performed a homology-based comparison to predict the DEs’ active site and identify a potential inhibitor. Each modeled DE was submitted to the DALI server to identify the closest homolog with protein structure information [31]. The DE model was then superimposed on its homolog structure obtained from DALI (hereafter referred to as template) using the Chimera (v. 1.17.3) Matchmaker function, and the interaction site was visualized via the Discovery Studio Visualizer (v24.1.0.23298) [32]. The CB-Dock2 web server (https://cadd.labshare.cn/cb-dock2/index.php, accessed on 10 June 2024) was used to perform molecular docking between the modeled DEs and the potential inhibitor.

## 3. Results

The adult gut transcriptome data were uploaded under the NCBI SRA accession numbers SRR27695095, SRR27695096, SRR27695095, SRR27695094, and SRR27695093. For the larval data, we used a recently published transcriptome (SRR926618). The de novo transcriptomes were assembled from the *R. ferrugineus* laboratory-reared adult male’s gut, laboratory-reared adult female’s gut, field-caught female’s abdomen, and laboratory-reared female’s abdomen (Appendix A). The raw reads were generated for each assembled transcriptome: 79,548,899 for the RPW larva, 26,779,316 for the field-collected male’s gut, and 215,865,354 for the field-collected female’s gut. The total number of clean reads was 40,512,028 for the RPW larva and 26,765,101 for the male gut collected from the field, which yielded 31,312 contigs with an average length of 772 bp and an N50 length of 1305 bp. Likewise, 215,734,706 clean reads were generated for the female gut collected from the field, which yielded 67,747 contigs with an average length of 663 bp and an N50 length of 1018 bp (Appendix A).

### 3.1. Digestive Enzyme (DE) Transcriptome Profiling and Abundance

#### 3.1.1. Glycosidase Gene Expression

By excluding transcripts showing similarity to proteins from plants, bacteria, and fungi, we identified 34 glycosidases across adult and larva *R. ferrugineus* (Appendix A) (named using the format: Rfer–enzyme type–number). Of the 18 glycosidases showing higher expression in the larvae, 12 were found only in the larvae. The larva-specific glycosidases primarily consisted of the exoglucanase β glycoside hydrolase (GH) family 48 (GH48), family 45 (GH45), and family 31 (GH31), and alpha-amylases. In comparison, 16 glycosidases were expressed more in the adults, with 7 expressed exclusively in the adults. The seven adult-specific glycosidases were identified as α-amylases, α-glucosidase, endo-beta-1,4-glucanase, GH31, and GH48s. However, none of these exclusively expressed enzyme types were found only in the larvae or adults. For example, the alpha-amylase RferGly22 was expressed in both larvae and adults, despite there being different α-amylases exclusively expressed in larvae (RferGly28 and 34) and adults (RferGly16 and 20).

The average glycosidase expression of adults and larvae was 2391 TPM and 3652 TPM, respectively (Appendix A). The three most highly expressed RferGlys in the larva sample were RferGly3 (myogenesis-regulating glycosidase, MYORG) with 2896 TPM, RferGly18 (glucosidase 2 subunit beta) with 443 TPM, and RferGly17 (Mannosyl-oligosaccharide glucosidase) with 294 TPM. Notably, RferGly3 expression was significantly higher than that of the rest of the glycosidases. The three most highly expressed adult RferGlys were RferGly1 (GH48) with 1176 TPM, RferGly11 (myogenesis-regulating glycosidase) with 958 TPM, and RferGly10 (endoglucanase) with 548 TPM.

The RferGly heatmap shows RferGly3 forming a distinct group from the rest of the RferGlys (Appendix A). In addition, the second and third most highly expressed larva RferGlys, RferGly17 and 18, also formed a distinct cluster. In contrast, the three most highly expressed adult RferGlys, RferGly1, 10, and 11, formed a distinct cluster below RferGly3.

#### 3.1.2. Lipase Gene Expression

We identified 85 lipases in *R. ferrugineus* from the BLASTx query (Appendix A). Thirty-two lipases were more highly expressed in the larval samples, with 19 lipases expressed exclusively in the larvae. These 19 lipases included lipase 3s, pancreatic lipases, carboxylesterases, glycerol-3-phosphate dehydrogenase, and phospholipase A2s. In comparison, 53 lipases were expressed at higher levels in adults, with 34 expressed exclusively in adults. The 34 adult-specific lipases were predominantly lipase 1 and 3, pancreatic lipases, and phospholipases. None of the lipase types expressed exclusively in larvae or adults were unique to one life stage; lipase 1s, 3s, pancreatic lipases, carboxylesterases, glycerol-3-phosphate dehydrogenase, and phospholipases were expressed in both larvae and adults. Additionally, adults exhibited lower average overall lipase expression, with 3333 TPM compared to 5607 TPM in larvae.

The top three most highly expressed larva lipases were RferLip72 (pancreatic lipase-related protein 2), RferLip3 (phospholipase D2, PLD2), and RferLip27 (lysophosphatidylserine lipase ABHD12-like) (Appendix A). RferLip72 expression was notably high, with 3974 TPM, compared to 647 TPM and 238 TPM for RferLip3 and RferLip27, respectively. Moreover, RferLip72 expression was not found in adults. The RferLip heatmap shows Rfer72 forming a single cluster (Appendix A). Similarly, RferLip3 also formed a cluster of its own. Conversely, RferLip27 formed a cluster with RferLip83 (carboxylesterase COEA16).

The top three most highly expressed adult lipases were RferLip28 (lipase 1-like), RferLip65 (60 kDA lysophospholipase), and RferLip47 (phospholipase A1-like), with 1102 TPM, 826 TPM, and 641 TPM, respectively. Additionally, RferLip28 and RferLip47 were not expressed in the larvae. These three highly expressed lipases formed a cluster in the lipase expression heatmap (Appendix A).

#### 3.1.3. Protease Gene Expression

201 proteases were identified across the larva and adult *R. ferrugineus*. Adults expressed more abundant proteases compared to larvae, with adults expressing an averaged total of 10,269 TPM proteases, as opposed to 9421 TPM in larvae (Appendix A). The three proteases with the highest expression in the larva sample were RferPro27 (aspartic proteinase), RferPro22 (venom dipeptidyl peptidase 4), and RferPro96 (xaa-Pro aminopeptidase ApepP-like) with 1530 TPM, 1026 TPM, and 956 TPM, respectively. Additionally, we observed that the protease expression heatmap shows that these three proteases were clustered together (Appendix A).

In contrast, the three proteases with the highest expression within the adult sample were RferPro46 (zinc carboxypeptidase A 1), RferPro101 (zinc carboxypeptidase A 1), and RferPro89 (trypsin-like serine protease) with 1537 TPM, 1367 TPM, and 1270 TPM, respectively. On the protease expression heatmap, RferPro46, 101, and 89 were clustered together with RferPro47 (glandular kallikrein-like) and RferPro30 (granzyme G-like protein).

Of the 201 proteases, 53 were more highly expressed in the larva sample, with 26 not expressed in the adult sample. These 26 larva-exclusive proteases primarily consisted of trypsins, chymotrypsins, cysteine proteases, zinc carboxypeptidases, and aminopeptidases. Conversely, 148 proteases were expressed more by adult *R. ferrugineus*, with 89 being adult-specific. Adult-specific proteases included trypsins, chymotrypsins, carboxypeptidases, aminopeptidases, and aspartic proteases. However, in comparison of the adult- and larva-specific proteases, the adult sample had notably more diversity of venom serine proteases, carboxypeptidases, and aminopeptidases.

### 3.2. DE Phylogenetic Analysis

#### 3.2.1. Glycosidase

To elucidate the relationships among the glycosidases, we constructed a phylogenetic tree consisting of glycosidases from the rice weevil (*Sitophilus oryzae*), mountain pine beetle (*Dendroctonus ponderosae*), Colorado potato beetle (*Leptinotarsa decemlineata*), and Asian long-horned beetle (*Anoplophora glabripennis*), with the silkmoth (*Bombyx mori*) as the outgroup (Figure 1). RferGly3 formed a clade with RferGly25 and shared the closest similarity with *S. oryzae* XP_030766422 (myogenesis-regulating glycosidase) (Figure 1). RferGly17 formed a distinct clade that includes an RferGly from *S. oryzae*, *D. ponderosae*, *L. decemlineata*, and *A. glabripennis* but no other RferGlys. RferGly18 also exhibited more similarity with some glycosidases from the other coleopterans and *B. mori* than other RferGlys. It shares the closest similarity with RferGly19 (glucosidase 2 subunit α). In contrast, RferGly1 shared the closest similarity with RferGly24 (GH48) and 31 (endoglucanase B) among the *R. ferrugineus* glycosidases, followed by *S. oryzae* ADU33252 (GH48). RferGly10 shared the closest similarity with RferGly27 (endoglucanase) and 9 (endo-*β*-1,4-glucanase), followed by *D. ponderosae* XP_019754618 (endoglucanase). RferGly11, however, formed a distinct clade that includes other coleopterans and *B. mori* but no other RferGlys. The most similar RferGly to RferGly11 was RferGly8 (GH31).

#### 3.2.2. Lipase

The phylogenetic analysis of RferLip displayed very complex inter-relationships between the coleopteran lipases (Appendix A). The three most highly expressed larva and adult lipases were very distinct from one another based on the fact that each formed distinct clades from the others. For the larvae, RferLip3 formed a distinct clade, with *S. oryzae* XP_030747183 (phospholipase D2) being the closest member of the clade (Figure 2a). It shared the closest similarity with RferLip6 (phospholipase DDHD2), which belongs to the neighboring clade. RferLip72 formed a clade with RferLip35 (pancreatic lipase 2-like), and along with other coleopterans, primarily *S. oryzae* and *A. glabripennis* (Figure 2b). RferLip27 formed a clade with *S. oryzae* XP030761237 and *D. ponderosae* XP019759955 (Figure 2b).

The three highest lipase expressions in the adult *R. ferrugineus* each formed a distinct clade from one another formed distinct clades from one another. RferLip28 formed a clade with RferLip36 (lipase 1-like) and *S. oryzae* and *D. ponderosae* lipases, whereas RferLip65 formed a clade with RferLip51 (Figure 2a). Rfer27 formed a clade with RferLip15 (lysophosphatidylserine lipase ABHD12-like), and RferLip65 shared the closest similarity with RferLip77, along with *S. oryzae* ANS53401 (esterase 1) and XP_030768230 (carboxylesterase 5A-like) (Figure 2b). Conversely, RferLip47 formed a distinct clade of its own. RferLip47′s neighboring clade consisted of 14 *R. ferrugineus* lipases.

#### 3.2.3. Aspartic Protease

Phylogenetic analysis for proteases was restricted to only aspartic proteases; the most highly expressed larval protease, RferPro27, is a member. This is due to the limitation of producing a massive phylogenetic tree of *R. ferrugineus* proteases along with its coleopteran evolutionary relatives. Only four *R. ferrugineus* aspartic proteases were found across the larva and adult samples. RferPro27 formed a distinct clade that branched off early in the phylogenetic tree, consisting only of itself and *S. oryzae* XP_030767631.1 (lysosomal aspartic protease) (Figure 3). Notably, each *R. ferrugineus* protease formed a distinct clade from the others, sharing more similarities with other coleopterans’ aspartic proteases. Only RferPro20 (cysteine protease ATG48) formed a distinct clade consisting only of itself.

### 3.3. Genome-Wide Analysis of the Candidate Digestive Enzymes

The genome-wide analysis of RferGly1 to RferGly34 systematically identified the gene distribution in the *R. ferrugineus* genome and ultimately identified the scaffold distribution of RferGLy genes and their locus_tag identifiers (Appendix A). We were more interested in tracing the tandem duplicated RferGly genes and alleles, which were identified using criteria to determine whether they were distributed in the same scaffold or a different scaffold. The genomic organization of RferGlys revealed that they were distributed across different scaffolds in the *R. ferrugineus* genome with an uneven distribution pattern (Appendix A). Using NCBI DBSOURCE accession numbers JAACXV010014020.1, JAACXV010014410.1, and JAACXV010000235.1, we annotated a deduced amino acid sequence of the three distinct RferGly genes. Three scaffolds were identified as carrying at least one, and often more than one, RferGly gene. The first RferGly gene in scaffold_65774 (RferGly1, RferGly24, and RferGly31) has four tandem duplicates (NCBI locus_tag ID = “GWI33_016026, GWI33_016027, GWI33_016029, and GWI33_016030”) (JAACXV010014020.1) (Appendix A). The second RferGly gene in scaffold_66208 (RferGly5, 6, 7, 26, and 29) has three tandem duplicates (NCBI locus_tag ID = “GWI33_019246, GWI33_019247, and GWI33_019249”) (JAACXV010014410.1). The third RferGly gene in scaffold_236 (RferGly20, 22, 28, and 34) has four tandem duplicates (NCBI locus_tag ID = “GWI33_016130, GWI33_016135, GWI33_016136, and GWI33_016137”) (JAACXV010000235.1) (Appendix A). We found two closely related genes distributed in different scaffolds. We found RferGly10 distributed in scaffold_117462 (JAACXV010018750.1) within the locus_tag GWI33_000901, which shared 92% amino acid identity with an orthologous sequence in the locus_tag GWI33_000903 of scaffold_117463 (JAACXV010018751.1) (Appendix A).

The genomic organization of RferLip family genes revealed that they were distributed across different scaffolds in the *R. ferrugineus* genome (Appendix A). Using NCBI DBSOURCE accession numbers, we annotated a deduced amino acid sequence of the thirteen distinct RferLip genes, which predominately emerged through tandem duplication. The respective scaffolds were identified as carrying at least one, and often more than one, RferLip gene in different locus_tags in the same scaffolds. The RferLip in scaffold_65771 contained two distinct genes. It was mapped in the locus tags GWI33_015970 and GWI33_015972 in the *R. ferrugineus* genome, indicating possible emergence through tandem duplication (Appendix A). RferLip11, RferLip16, and RferLip25 were represented in the locus tags GWI33_004295, GWI33_004299, and GWI33_004300, respectively, and were distributed across the same scaffold, scaffold_228. scaffold_66285 had twelve distinct tandem duplicates that were distributed among the NCBI locus_tag IDs, “GWI33_019719, GWI33_020600, GWI33_007629, GWI33_007628, GWI33_020602, GWI33_020603, GWI33_020601, GWI33_020604, GWI33_020606, GWI33_019723, GWI33_019722, and GWI33_007630”. scaffold_66394 had seven distinct tandem duplicates that were distributed among the locus_tags “GWI33_020604, GWI33_020606, GWI33_020601, GWI33_0206042, GWI33_020602, GWI33_020603, and GWI33_0206042”. We identified three distinct tandem duplicates in the same scaffold, scaffold_71, that were distributed in the locus_tags GWI33_021631, GWI33_021632, and GWI33_021636 (Appendix A). Three distinct tandem duplicates were identified in scaffold_376 and were distributed in the locus_tags GWI33_007628, GWI33_007629, and GWI33_007620. Each of scaffold_17, scaffold_271, and scaffold_407 possesses two distinct tandem duplicates, with their locus identifiers given in Appendix A. The remaining four RferLip genes were distributed in different scaffolds and mapped in the locus tags, and these RferLip genes were all distinct genes with no evidence of duplication (Appendix A).

We identified a total of 11 different allelic variants of RferLip, viz., RferLip4, RferLip5, RferLip10, RferLip13, RferLip30, RferLip34, RferLip46, RferLip60, RferLip61, RferLip68, and RferLip82, at a specific location (locus_tags) on a chromosome (scaffold) (Appendix A). The amino acid sequences of the allelic variants of RferLip4, RferLip13, RferLip30, RferLip46, RferLip68, and RferLip82 have >98% identity (Appendix A). We identified the variations between alleles through indels (RferLip4, 5, 10, and 61), single-site substitutions (RferLip13, 30, and 46), and complete repeat indels (RferLip34, 68, and 82). The functional RferLip gene length, CDS length, percentage of amino acid identity, and protein alignment are shown in Appendix A.

We retrieved 35 scaffolds in the *R. ferrugineus* genome that were identified as carrying at least one, and often more than one, RferPro gene using NCBI DBSOURCE accession numbers (Appendix A). RferPro9, RferPro70, RferPro102, and RferPro134 were, respectively, in scaffolds_44, 30, 66088, and 0, and each contained four distinct tandem duplicates and is mapped in different locus tags as shown in Appendix A. RferPro5, RferPro34, RferPro40, RferPro46, RferPro79, RferPro97, RferPro101, RferPro105, RferPro148, RferPro175, RferPro178, RferPro188, RferPro199, and RferPro200, in their respective scaffolds with locus tag identifiers, contained three distinct tandem duplicates (Appendix A). Each of scaffolds_66422, 66403, 165, 50, 0, 116786, 117862, 66360, 66233, 355, 77, 65689, 117284, 66360, 28, and 21 possesses two distinct RferPro tandem duplicates, with their locus identifiers given in Appendix A. The remainder of the RferPro enzymes were distributed in the scaffolds, and each member was mapped in single locus tag identifiers, with no evidence of a duplication event.

We identified a total of 21 allelic variants of RferPro, viz., RferPro4, RferPro10, RferPro16, RferPro33, RferPro38, RferPro42, RferPro47, RferPro56, RferPro70, RferPro80, RferPro105, RferPro115, RferPro117, RferPro119, RferPro120, RferPro123, RferPro128, RferPro132, RferPro140, RferPro177, and RferPro193, at a specific location (locus_tags) on a chromosome (scaffold) (Appendix A). Each allelic variant’s amino acid sequence has >97% identity (Appendix A). We identified the variations between alleles through indels (RferPro4, 10, 16, 38, 42, 115, 117, 132, 140, 177, and 193), single-site substitutions (RferPro47, 80, 105, 120, 123, and 128), and complete repeat indels (RferPro33, 56, 70, and 119). The amino acid sequence alignment of the different RferPro alleles sharing over ≥97% identity is given in Appendix A.

### 3.4. Modeling Evaluation and Molecular Docking Analysis

#### 3.4.1. Glycosidase Candidate Model and Putative Inhibitor Docking

RferGly3 was chosen for modeling and subsequent docking analysis due to it having the highest expression level among larva glycosidases while also being expressed by the adult *R. ferrugineus*. The modeled RferGly3 had a predicted template modeling (pTM) score of 0.94, which indicates very high accuracy (the maximum pTM score is 1.0). RferGly1 had an ERRAT score of 98.363 (Table 1). In ERRAT evaluation, a score greater than 50 indicates a good model [27]. Similarly, the Verify3D score acquired was 88.15%, which surpassed the threshold of 80% for a model with good agreement [28]. MolProbity analysis also revealed that 94.77% of the RferGly3 residues fall within the favorable region. Only 0.16% of residues fall within the outlier region. The RferGly3 QMeanDisco Global score was 0.8, which implied that the structure has high agreement regarding its structural features and energetics [29]. In addition, the calculated MolProbity score was 1.67, which indicates minimal steric clashes within the structure.

In a DALI structural similarity search, RferGly1 shared a degree of structural similarity with a *B. mori* glucoside hydrolase (PDB ID: 6LGA; hereafter named 6LGA). RferGly1 had a structural difference with a root mean square deviation (RMSD) of 1.16 Å with 6LGA, indicative of high similarity (Table 1). We performed a similar evaluation of the 6LGA structure as an additional benchmark for RferGly3 model quality. Only RferGly3’s ERRAT score was better than that of 6LGA, whereas 6LGA had better model assessment scores on all other parameters. Nevertheless, RferGly3 is a reliable and accurate model based on the evaluated parameters. The SMART scan reported a GH31 domain within the RferGly3 sequence, which identified RferGly3 as a member of GH31 (Appendix A).

To evaluate the binding activity of this predicted active site, we performed molecular docking via CB-DOCK2 of RferGly3 and 6LGA with known glycoside inhibitors, 1,4-dideoxy-1,4-imino-D-arabinitol (DAB) and 1-deoxynojirimycin [33]. CB-DOCK2 revealed a RferGly1 cavity with a 670 Å^3^ cavity volume. This cavity contains, the predicted inhibitor position on RferGly1 that was in close proximity to the reported DAB and 1-deoxynojirimycin position bound to 6LGA (Appendix A; Figure 4a,b) [33]. Information on the residues of the cavity is summarized in Appendix A. The high degree overlap of DAB and 1-deoxynojirimycin docking position on RferGly1 supports the reliability of the predicted cavity as a plausible ligand-binding site (Figure 4c).

The interaction between each inhibitor and RferGly1 was visualized in a 2-dimensional (2D) diagram. Four conventional hydrogen bonds and six carbon–hydrogen bonds were formed between RferGly1 and DAB, with an additional seven residues exerting van der Waals forces (Figure 5a). The conventional hydrogen bonds were formed with Lys384, Asp443, and Asp276, whereas the carbon–hydrogen bonds were formed with Asn480, Asp276, Asp386, and Trp313. In contrast, RferGly1 and 1-deoxynojirimycin had three conventional hydrogen bonds, four carbon–hydrogen bonds, a single pi–sigma bond, and three unfavorable interactions that arose from steric clashes between the two molecules (Figure 5b). In addition, 10 RferGly1 residues exerted van der Waals forces on 1-deoxynojirimycin, excluding Trp244, which exerted a pi–sigma bond. The conventional hydrogen bonds were formed with Asp276, Asp386, and Asp443, and these three residues also formed carbon–hydrogen bonds with 1-deoxynojirimycin. Additionally, the unfavorable interactions occurred on Met476, Asp443, and Arg440.

#### 3.4.2. Lipase Candidate Model and Putative Inhibitor Docking

RferLip3 was selected for the modeling and docking analysis, given that it is the lipase most highly expressed by larvae, which is also expressed in adults (Appendix A). The RferLip3 model had a pTM score of 0.82, which was still within the region of good model accuracy (Table 1). The ERRAT score of RferLip3 was 91.56, which was also indicative of a good model. The Ramachandran plot analysis revealed that 93.88% of RferLip3 residues were in the favorable regions, with 0.81% falling in the outlier region. Furthermore, RferLip3 had a MolProbity score of 1.67, indicative of minimal steric clashes within the structure. However, the Verify3D and QMeanDisco Global scores were 73.35% and 0.6, respectively, which suggest moderate agreement with the expected structural patterns.

In the DALI structural search, the Gram-negative bacteria *Pseudomonas aeruginosa* phospholipase D (PDB ID: 7V55; hereafter named 7V55) shared high structural similarity with RferLip3 and, thus, was selected as the reference protein. The 7V55 had a structural difference of 2.2 Å from our RferLip3. It had better assessment scores across all the evaluated parameters. Nevertheless, the RferLip3 model exhibited good overall structural accuracy. To further verify the identity of RferLip3 as a PLD2, we performed motif searching for the tandem phox homology (PX), pleckstrin (PH), and two HKD domains characterized by HxKx_4_Dx_6_G(G/S), where x denotes amino acids between histidine lysine and aspartic residues [34]. The SMART scan validated that the RferLip3 sequence possesses all the domains associated with phospholipase D2 (Appendix A).

The study that reported the 7V55 structure also described its active site, which will be used as a reference for our RferLip3 and molecular docking [35]. Two known phospholipase D inhibitors, quercetin (PubChem CID: 5280343) and benzimidazolinone (PubChem CID: 11985), were selected for molecular docking analysis [36,37]. RferLip3 and 7V55 had a structural difference in terms of a root mean square differentiation (RMSD) of 2.2 Å. The RferLip3 structure was larger than 7V55 (Figure 6a,b). In the CB-DOCK2 molecular docking analysis, one of the predicted ligand cavities, 3334 Å^3^ in cavity volume, had inhibitors docked in very close proximity to reference 7V55′s docked inhibitors with some overlaps for the predicted benzimidazolinone docked position (Appendix A; Figure 6c).

In the visualized 2D interaction between RferLip3 and quercetin, three residues, Arg518, Gln816, and His931, formed four conventional hydrogen bonds with quercetin (Figure 7a). In addition, a single pi–pi interaction formed with Phe862, two unfavorable interactions formed on Asp959 and Arg952, and 11 residues exerted van der Waals forces on quercetin. Conversely, fewer interactions were formed between RferLip3 and benzimidazolinone. Asp687 and Gly861 formed two conventional hydrogen bonds with benzimidazolinone (Figure 7b). Asp687 also formed pi–anion interactions with the inhibitors. Two pi–pi interactions were formed with Phe683 and Trp413, and nine residues exerted van der Waals forces on the inhibitor.

#### 3.4.3. Protease Candidate Model and Putative Inhibitor Docking

The aspartic protease RferPro27 was selected for modeling and docking analysis as this protease was most abundantly expressed in larvae while also being expressed in the adult sample. The RferPro27 model had a pTM score of 0.85, which is still within the region of good model accuracy (Table 1). The ERRAT score of RferPro27 was 88.52, and the Verify3D analysis showed an 88.15% score. The Ramachandran plot analysis revealed that 95.64% of RferPro27 residues are in favorable regions, with 0.27% falling within the outlier region. Furthermore, RferPro27 had a MolProbity score of 1.85, indicating minimal steric clashes within the structure. Additionally, the QMeanDisco Global score was 0.75.

In the DALI structural search, the castor bean tick (*Ixodes ricinus*) cathepsin D (PDB ID: 5N7Q; hereafter named 5N7Q) had the most similar structure to RferPro27 and, thus, was selected as the reference protein. The 5N7Q had a structural difference of 0.675 Å from RferPro27, suggesting minor structural differences. It had better assessment scores across all the evaluated parameters; however, RferPro27’s overall scores were still indicative of a good model. The SMART scan reported the presence of the TAXi_N domain and aspartyl protease domain, which further verified the status of RferPro27 as an aspartic protease (Appendix A).

The study that reported the 5N7Q structure also described its active site, which will be used as a reference for our RferPro27 ligand-binding cavity prediction and molecular docking [38]. Two known aspartic protease inhibitors, pepstatin A and ellagic acid (PubChem CID: 5281855), were selected for molecular docking analysis [38,39]. RferPro27 and 5N7Q were similarly sized. CB-DOCK2 predicted an RferPro27 ligand cavity, 1751 Å^3^ in volume, that falls within the same region of the active site within 5N7Q (Appendix A; Figure 8a,b). The predicted RferPro27 ligand inhibitors’ docked positions showed significant overlap with the reference.

RferPro27 formed ten conventional hydrogen bonds, two carbon–hydrogen bonds, and four alkyl bonds with pepstatin A (Figure 9a). The conventional hydrogen bonds were formed with Gly88, Thr130, Gly132, Ser133, Asp272, Tyr247, Gly274, and Ser276. Carbon–hydrogen bonds were formed with Gly88 and Thr275, whereas alkyl–alkyl bonds were formed with Tyr131, Phe168, Met339, and Ala334. Additionally, 15 residues exerted van der Waals forces on pepstatin A. In contrast, interaction with ellagic acid formed four conventional hydrogen bonds, three carbon–hydrogen bonds, two pi–anion bonds on Asp272, a single pi–sulfur bond on Met330, and a single pi-stacked interaction on Gly132 (Figure 9b). The conventional hydrogen bonds were formed with Ser133, Gly274, and Met330, whereas the carbon–hydrogen bonds were formed with Gly132 and Thr275. In addition, eight residues exerted van der Waals forces on ellagic acid.

## 4. Discussion

An insect possesses a diverse arsenal of DEs to ensure efficient digestion. The DEs are synthesized in and secreted from the insect gut. The larva and adult *R. ferrugineus* gut transcriptomes were profiled for DEs. The DEs were categorized into glycosidases, which involve carbohydrate digestion; lipases, which involve protein digestion; and proteases, which involve protein digestion. The larvae and adults showed differential gene expression profiles, hinting at their respective stages’ dietary needs. The larvae expressed more gene transcripts encoding glycosidases and lipases, reflecting their primary goal of maximizing growth, whereas the adults expressed more proteases to meet the molecular demand for carrying out more complex activities such as egg production, hormone secretion, deterring mating competitors, and molting [10].

This study focused on the three most highly expressed glycosidases, lipases, and proteases within the larva and adult *R. ferrugineus*. The phylogenetic analysis revealed that the top three DEs with the highest expressions belonged to different clades. The most highly expressed DEs among glycosidases, lipases, and proteases expressed in both larvae and adults were RferGly3, RferLip3, and RferPro27, which were modeled and characterized. We prioritized larva expression because the larva inflicts the most damage on the host palm tree in the *R. ferrugineus* lifespan [10]. RferGly3 and RferLip3 showed similar ligand-binding cavity structures to their respective reference proteins. However, RferPro27, which had the smallest difference from its reference protein compared to RferGly3 and RferLip3, exhibited a significantly different ligand docked position for its tested inhibitors, ellagic acid, and pepstatin A.

The RferDEs were not docked with their substrates due to the infeasibility of performing molecular docking on large macromolecules of oligosaccharides, phospholipids, and peptides. As an alternative, the RferDEs were docked with their corresponding competitive inhibitors that are known to bind to the enzyme’s active site. Two inhibitors were used for each RferDE so that a consensus ligand-binding cavity of the RferDE could be identified.

Glycosidase MYORG refers to a family of transmembrane proteins associated with the endoplasmic reticulum, the Golgi apparatus, or the nuclear envelope membranes involved with the development of musculature and animal movements [40]. MYORG can belong to GH31 and GH family 13 (GH13) glycosidases, which are among those involved in myogenesis regulation [41]. The SMART scan verified the status of RferGly3 as a member of GH31 and not GH13. However, the DALI analysis revealed that a GH13 sucrose hydrolase shared the closest structural similarity with RferGly3. This is likely due to the similarities of how both GH families acting on the alpha-glucosidic linkage [42]. Thus, they share many structural commonalities in order to bind to similar glycoside linkages. Furthermore, both families are involved with myogenesis regulation, glycogen metabolism, amino acid transport, and quality control of N-glycosylation in the endoplasmic reticulum [41].

GH31 has diverse members of different specificities. However, a commonality within GH31 is that these glycosidases’ substrates consist of oligosaccharides. In myogenesis, GH31 can break down oligosaccharides like sucrose to produce simpler sugars, glucose, and fructose. These monosaccharides fuel ATP production via Krebs’ cycle, providing the cells with energy for building muscle fibers and assisting in movements [43]. In the context of larvae, myogenesis is very important for enabling movement and growth.

In contrast, GH48, which is a cellulase, was the most highly expressed glycosidase in adults. Cellulase is vital for digesting the cell wall of sugarcane fibers, maximizing polysaccharide sources. For adult *R. ferrugineus*, myogenesis is necessary for the maintenance of its muscle and for enabling movement. Therefore, myogenesis-regulating glycosidases may not be as important as in the larvae. We compared the total TPM for myogenesis-regulating glycosidase, including GH13, from the glycosidase expression profiles between larvae and adults (Appendix A). The abundance of these enzymes was 3073 TPM for the larvae and 899 TPM for the adults. This further emphasized the inference that myogenesis is a process less essential for adult *R. ferrugineus*.

Moreover, recent research found MYORG’s significance in ensuring proper neurodevelopment. The mutation of MYORG has led to primary familial brain calcification, wherein various brain regions exhibited abnormal calcium depositions linked to psychiatric conditions, motor impairments, and cognitive decline [44,45]. Therefore, RferGly3 MYORG may be vital for the proper neurodevelopment of *R. ferrugineus* larvae and muscle development.

The selected inhibitors, DAB and 1-deoxynojirimycin, are known glycosidase inhibitors that bind to the enzyme’s active site [33]. 1-deoxynojirimycin and DAB are alpha-glucosidase inhibitors derived from the Mulberry plant for hyperglycemia treatment [46]. These compounds demonstrated toxicity against silkmoth larvae (*Samia ricini*), eastern subterranean termites (*Reticulitermes flavipes*), and cockroaches (*Periplaneta americana*) [47,48,49]. Their inhibitory effect could restrict glucose availability to *R. ferrugineus*, inducing an antifeeding effect. 1-deoxynojirimycin was modified to form N-linked dimers, which demonstrated selective toxicity against the insect’s trehalase [50]. This opens new avenues for manipulating digestion-related treatment drugs to develop novel, selective insecticides.

The lipase PLD’s primary molecular function is hydrolyzing phospholipids, such as phosphatidylcholine (PC), to release phosphatidic acid (PA) and also the transphosphatidylation of PC in the presence of primary alcohols [34,51]. During neurotransmission, PA docks and primes the neuron secretory vesicles and fusion [51]. Thus, the inhibition of PLD would lead to less production of PA, leading to impaired neurotransmission activity. Furthermore, it was demonstrated that the ablation of PLD led to delayed brain development and impaired cognitive activity, which was attributed to a decline in acetylcholine synthesis and choline-containing phospholipids such as sphingomyelin that make up the myelin sheath of the neuron [52]. Furthermore, PLD also mediates the cell proliferation process by activating the rapamycin protein’s mammalian target, associating itself with cell proliferation [53].

Therefore, inhibiting PLD would potentially disrupt digestive, neurotransmission, and neurodevelopment processes. On the other hand, the adults had phospholipase A1 (RferLip47), which was among the most highly expressed lipases. PLA1 cleaves the ester bond at the sn-1 position of phospholipids and releases fatty acids [51,54]. PLA1 members share a common GXSXG motif as their catalytic active site [54]. Free fatty acids also play an important role in neurotransmission [51]. The difference between the expression of PLD and PLA1 between the larvae and adults could be attributed to the more specific role of PA and FFAs. PLD’s role is very important in the early stage of brain development and in enhancing cell proliferation. Thus, high PLD expression is necessary for optimal larva development. Therefore, the inhibition of PLD would impair *R. ferrugineus* development and neuronal function, apart from inhibiting lipid digestion.

The reference lipase that shared close similarity to RferLip3 was also identified as a PLD. Quercetin and benzimidazolinone were selected for molecular docking analysis because both are broad-range phospholipase inhibitors. Quercetin is a flavonoid found across diverse plant species. It has demonstrated growth-inhibitory effects against *B. mori*, *Helicoverpa armiera*, and *Spodoptera exigua*, and multitudes of coleopterans, including *L. decemlineata* [55,56]. Benzimidazolinone is a derivative of benzimidazole, which demonstrated toxicity against the coleopteran *Tribolium castaneum* [37,57]. Benzimidazolinone and quercetin showed an overlapping docking position in both RferLip3 and 7V55 and an overlapping predicted ligand-binding cavity. However, the degree of overlap was lower than that of RferGly3 and its corresponding reference. This is probably due to 7V55 being a protein from a Gram-negative bacterium, which is very dissimilar from insects. However, the small degree of deviation between the two PLDs suggests the significance of their activities and that the structures were highly conserved. This could be attributed to PLDs’ diverse critical biological roles, as described earlier. Taken together, the previous research that reported toxicity against coleopterans and the observed similarity, and the predicted ligand-binding position of RferLip3 being very similar to that of 7V55, suggest that quercetin and benzimidazolinone can potentially induce mortality in *R. ferrugineus*.

Some proteases demonstrate additional roles beyond nutrient acquisition. In insects, a subset of proteases comprise the insect’s venom [58]. Certain members of serine proteases regulate the Toll pathway that controls the insect’s innate immune system [59]. Aspartic protease degrades intracellular and extracellular proteins in lysosomes as part of nutrient recycling. Aspartic proteases are grouped into 16 families based on their active site sequence similarity [60].

Apart from protein digestion, aspartic protease also plays a vital role in insect metamorphosis. The activation of lysosomes and aspartic protease activity, later identified as cathepsin D [61], mediated the replacement of the larva with pupal body fat. During the larval-to-pupal transition, cathepsin D induced the programmed cell death of larval fat body histolysis [62]. Therefore, the inhibition of cathepsin D would potentially induce development arrest that hinders proper pupa development. RferPro27 had a very small structural difference from its reference cathepsin D, 5N7Q, which support our prediction that RferPro27 could be a cathepsin D.

Zinc carboxypeptidase A1, which was the most highly expressed protease in adult R. ferrugineus, also primarily involved protein digestion, mediating metamorphosis via molting [63]. However, adult weevils do not undergo molting [64]. Carboxypeptidase A1 is likely a necessary protease for efficient protein digestion. The adult *R. ferrugineus* is required to expend energy for movement and reproduction.

Pepstatin A is a broad-range aspartic protease inhibitor isolated from actinomycetes [65]. This compound was researched for therapeutic effects in combating neurodegenerative disease, cancer, and HIV infection, with these diseases involving aspartic protease activity [66,67,68]. There are no specific studies that evaluated pepstatin A’s inhibitory effects on insects. However, pepstatin A was utilized in a validation test for characterizing an insect aspartic protease [38,69]. Ellagic acid is a type of polyphenol found in fruits, nuts, and seeds such as pomegranates, raspberries, and walnuts [70]. This compound was also studied for anticancer and antimicrobial properties [71,72]. Ellagic acid demonstrated a growth impairment effect on cotton leafworm (*Spodoptera litura*).

The predicted ligand-binding position significantly overlapped with the reference ligands’ positions. Pepstatin A and ellagic acid interact with aspartic protease on the active site. The similarity of the domain to the reference, the small RMSD score, the overlapping ligand-binding cavity, and the docked ligand positions affirm that RferPro27 is an aspartic protease with an active site similar to 5N7Q.

We manually annotated DEs in the *R. ferrugineus* genome, and these annotations allowed us to correct the transcriptome annotation. Genome (GenBank: GCA_014462685.1) analyses revealed several DE duplications from two to seven genes in clusters in the same scaffold. Several genes were predicted to be tandem duplicates and allelic variants among the glycosidase, lipase, and protease groups. Among the glycosidases, seven groups of splice variants were identified (Appendix A). Most notably, the glycosidases were mapped in scaffold 236 (alpha-amylase) and scaffold 66,208 (GH48), each consisting of five glycosidase transcripts. GH48 members are widely distributed across coleopterans, and they were inherited from a Phytophaga ancestor that likely acquired them via distinct horizontal gene transfer events from an Ascomycete fungus and a species of Actinobacteria, respectively [2,73]. Fourteen potential splice variants were identified from the lipase group (Appendix A). The synergetic collaboration between GH48, GH45, and GH9 is well documented in coleopterans [74]. A functional analysis of glycosidase would be particularly interesting, especially for the tandem duplicates, to elucidate the diverse roles they play in digestion. Scaffold_66285 (triacylglycerol lipase, TAG lipase) had the greatest number of potential tandem duplicates, with 13 RferLip transcripts mapped onto it. TAG lipases have an important role in hydrolyzing the outer ester links of TAGs [75], which are hypothesized to be active in weevil feeding and digestion. RferPro had the greatest number of tandem duplicates, with 42 groups (Appendix A). Among the RferPro members, scaffold_0 (JAACXV010000001.1) (venom serine carboxypeptidase-like) had the most duplicates, with 11 transcripts mapped onto it. Interestingly, we identified several allelic variants of RferLip and RferPro, which shared >95% identity, with most of them emerging through single-site mutation and small indels (Appendix A). The allelic variation in RferLip and RferPro genes is likely essential for these multifunctional digestive roles, accounting for the functional variations, representing an exciting area of further research.

## 5. Conclusions

Using its genome and transcriptome data, we comprehensively characterized the glycosidase, lipase, and protease families of the Asian palm weevil, *R. ferrugineus*, and identified allelic variants and duplicates. The transcriptome analysis of *R. ferrugineus* adults and larvae identified several important DEs for nutrient digestion, with additional roles in neurodevelopment and metamorphosis. RferGly3, RferLip3, and RferPro27 were highly expressed in the larvae, highlighting their significance for survival. Molecular models of these enzymes exhibited significant structural similarities with corresponding reference proteins. Additionally, the reference proteins were functionally similar to their corresponding RferDEs. Only RferGly3 differed in GH type from its reference. However, both shared many functional similarities, as described in previous research in the literature. Furthermore, the predicted inhibitor binding positions showed high similarity with the references’ ligands, suggesting comparable active sites.

Our study provides a profile of the DE secretions within the larva and adult *R. ferrugineus* guts and characterizes DEs that are potential targets for inhibition. However, the caveat of the approach of inhibiting digestive enzymes is that the inhibitors may not be effective on their own. Increasing the inhibitor dosage would incur additional costs and potentially endanger non-target species and human consumers. Therefore, the ideal approach would be to inhibit a combination of highly expressed DEs reported in this study to achieve effective pest control with minimal dosage. The insights from this study pave the way for developing novel rationally based insecticides or new formulations that can be investigated in future research.

## Figures and Tables

**Figure 1 insects-16-00421-f001:**
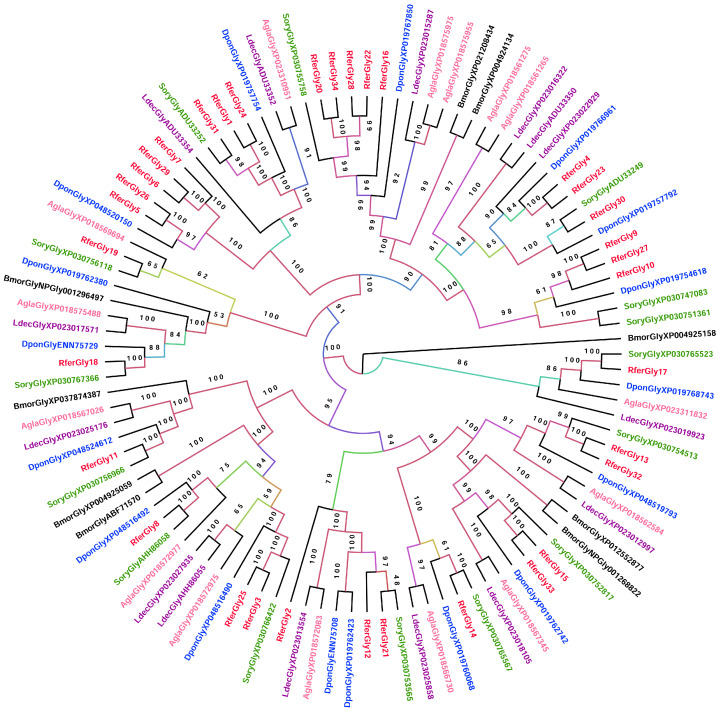
Maximum likelihood tree of glycosides identified from *R. ferrugineus* (Rfer) and other coleopterans from NCBI, including *Dendroctonus ponderosae* (Dpon), *Anoplophora glabripennis* (Agla), *Leptinotarsa decemlioneata* (Ldec), and *Sitophilus oryzae* (Sory). The tree is rooted with the *Bombyx mori* (Bmor). The numbers on the branches are bootstrap values (UFBoot *n* = 1000). Scale bar = 2 amino acid substitutions per site. The branches are colored based on the bootstrap values. NCBI accession nos. are included with each taxon.

**Figure 2 insects-16-00421-f002:**
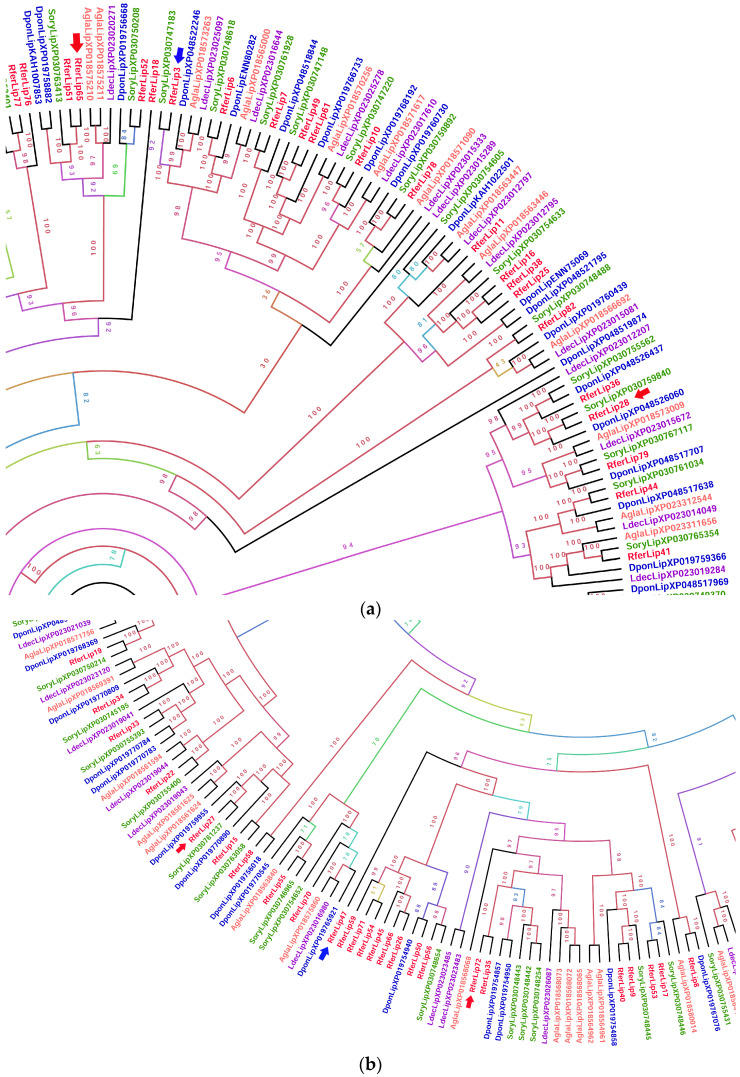
Cropped zoomed-in sections of the maximum likelihood unrooted tree of *R. ferrugineus* lipases (RferLip) (see Appendix A for the complete phylogeny). The tree was built from the alignment of the lipases sequences of *R. ferrugineus*, Rfer (red), *Anoplophora glabriipennis*, Agla (pink), *Dendroctonus ponderosae*, Dpon (blue), *Leptinotarsa decemlioneata*, Ldec (purple), and *Sitophilus oryzae*, Sory (green). The tree branch appearance was colored based on the bootstrap values. (**a**) Section highlighting RferLip3 and RferLip28. (**b**) Section highlighting RferLip27, 47, 72, and 35. Blue arrows indicate the most highly expressed lipase among the larva RferLip members, whereas the red arrows indicate the most highly expressed among the adult RferLip members. Scale = 2.0 amino acid substitutions per site. The full phylogenetic tree of RferLip is shown in Appendix A.

**Figure 3 insects-16-00421-f003:**
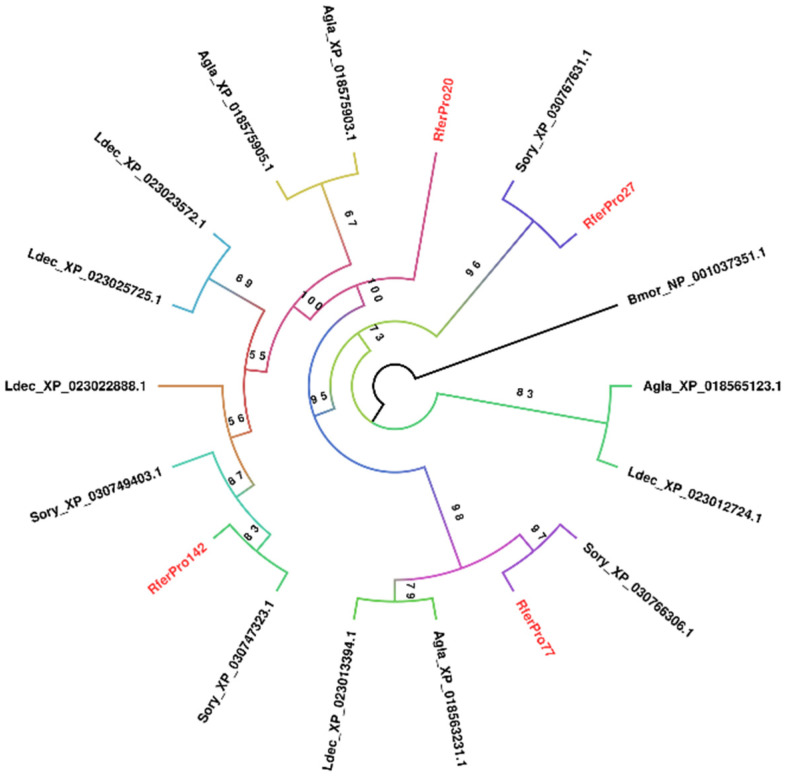
Maximum likelihood rooted tree of aspartic proteases predicted from *R. ferrugineus*, Rfer (red) and several other coleopterans. Includes species from *Anoplophora glabriipennis* (Agla), *Coccinella septempunctata* (Csep), *Dendroctonus ponderosae* (Dpon), *Leptinotarsa decemlioneata* (Ldec), and *Sitophilus oryzae* (Sory), with *B. mori* (Bmor) as an outgroup. The tree branch appearance was colored based on the bootstrap values. NCBI accession nos. are included with each taxon.

**Figure 4 insects-16-00421-f004:**
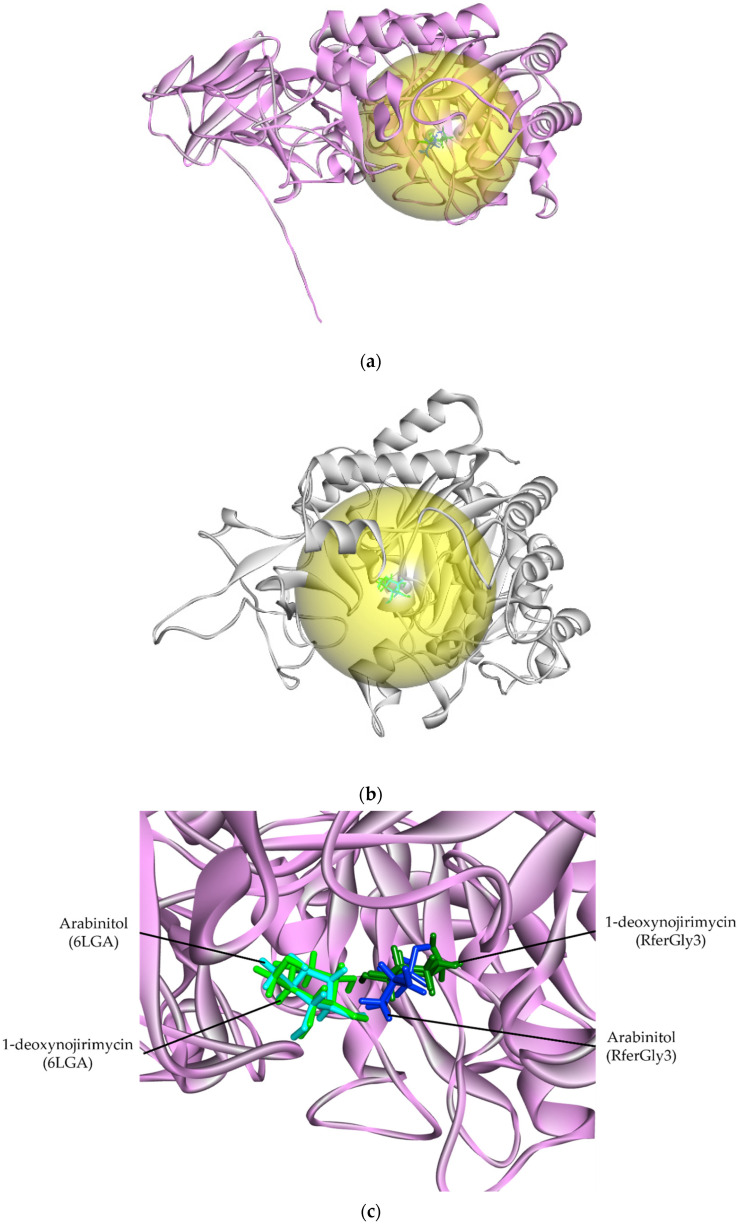
The ligand interaction cavity and the interaction of glycosidase inhibitors, 1,4-dideoxy-1,4-imino-D-arabinitol (DAB), and 1-deoxynojirimycin with RferGly1 and 6LGA. (**a**) The predicted ligand-binding cavity of RferGly3 (yellow); (**b**) the reported ligand-binding cavity of 6LGA (yellow); (**c**) the superimposed inhibitors’ docked positions on the RferGly3 structure. Dark blue: RferGly3 DAB; light blue: reference 6LGA DAB; dark green: RferGly3 1-deoxynojirimycin; light green: 6LGA 1-deoxynojirimycin.

**Figure 5 insects-16-00421-f005:**
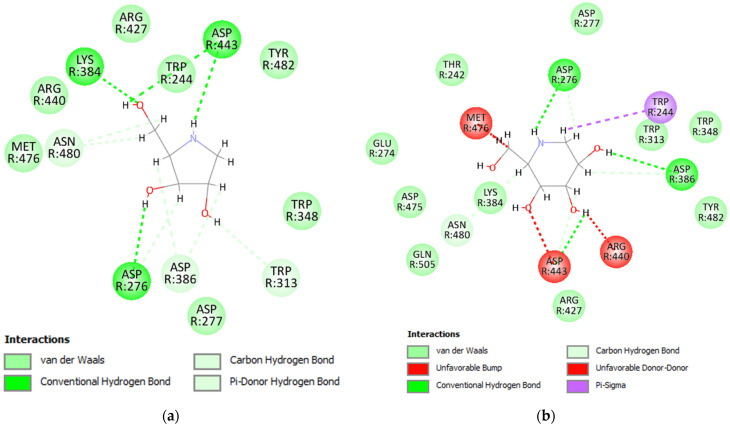
Two-dimensional (2D) interaction between RferGly3 protein and inhibitors. (**a**) Interaction between RferGly3 and DAB; (**b**) interaction between RferGly3 and 1-deoxynojirimycin.

**Figure 6 insects-16-00421-f006:**
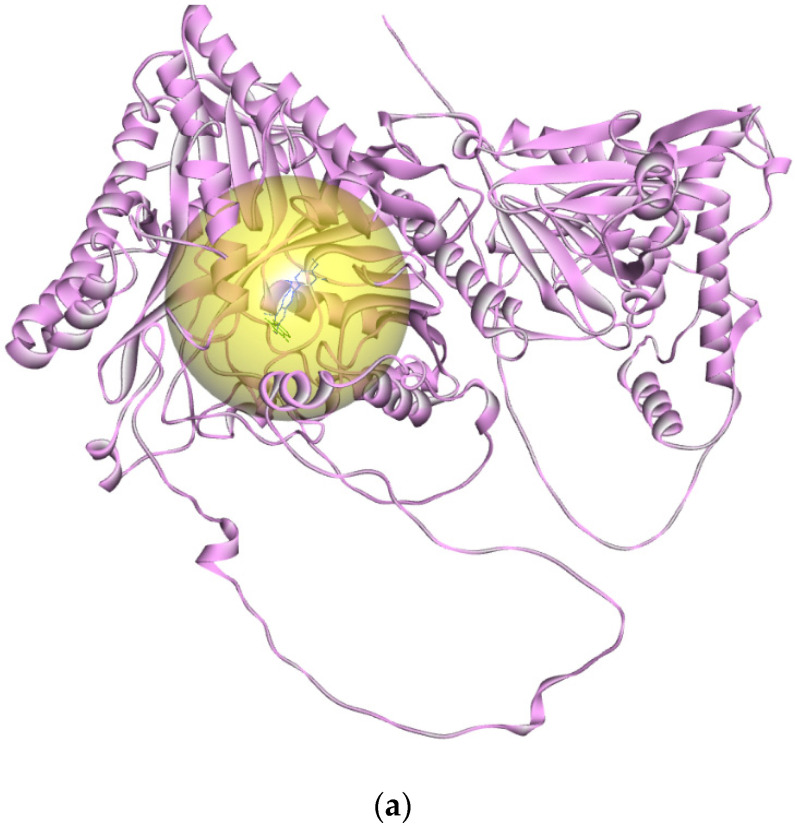
The ligand interaction cavity and the interaction of phospholipase D inhibitors, quercetin, and benzimidazolinone with RferLip3 and reference protein 7V55. (**a**) The predicted ligand-binding cavity of RferLip3 (yellow); (**b**) the reported ligand-binding cavity of 7V55 (yellow); (**c**) the superimposed quercetin and benzimidazolinone docking positions on the RferLip3 structure. Dark blue: RferLip3 quercetin; light blue: 7V55 quercetin; dark green: RferLip3 benzimidazolinone; light green: 7V55 benzimidazolinone.

**Figure 7 insects-16-00421-f007:**
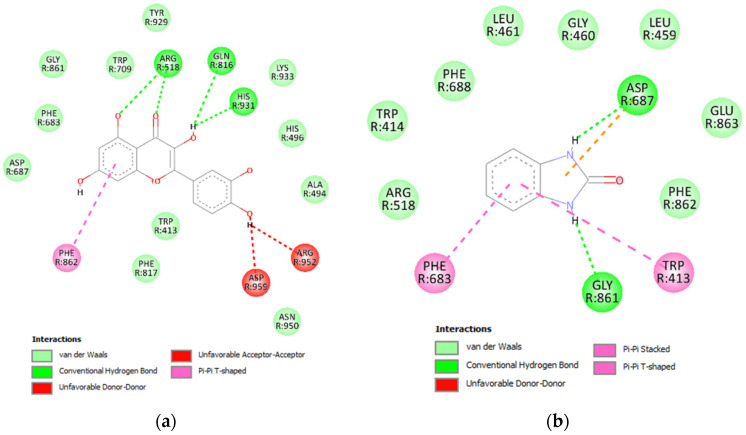
2D interaction between RferLip3 protein and its inhibitors, quercetin and benzimidazolinone. (**a**) Interaction between RferLip3 and quercetin; (**b**) interaction between RferGly3 and benzimidazolinone.

**Figure 8 insects-16-00421-f008:**
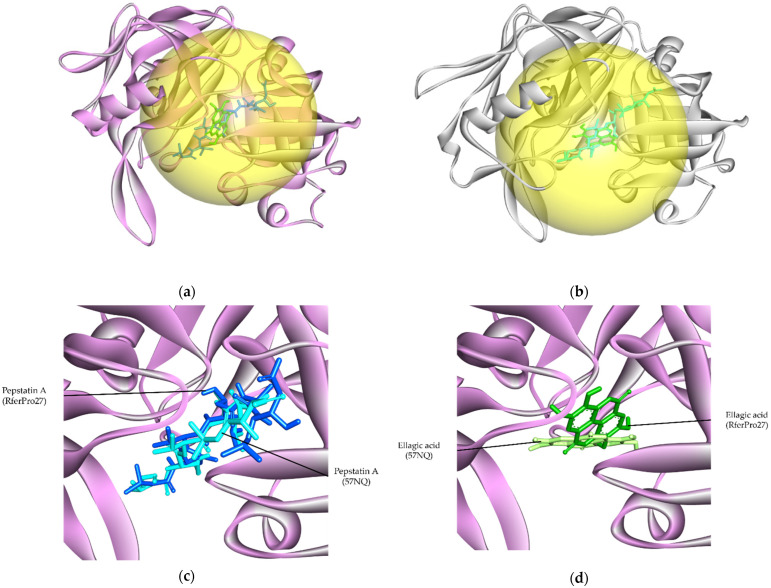
The ligand interaction cavity and the interaction of aspartic protease inhibitors, pepstatin A, and pelagic acid with RferPro27 and reference protein 5N7Q. (**a**) The predicted ligand-binding cavity of RferPro27 (yellow); (**b**) the reported ligand-binding cavity of 5N7Q (yellow); (**c**) the superimposed pepstatin A docked positions on the superimposed RferPro27 and 5N7Q structure; (**d**) the superimposed ellagic acid docked positions on the superimposed RferPro27 and 5N7Q structure. The interfering residues that restricted inhibitors’ access are highlighted in yellow. Dark blue: RferPro27 pepstatin A; light blue: 5N7Q pepstatin A; dark green: Rferpro27 ellagic acid; light green: 5N7Q ellagic acid.

**Figure 9 insects-16-00421-f009:**
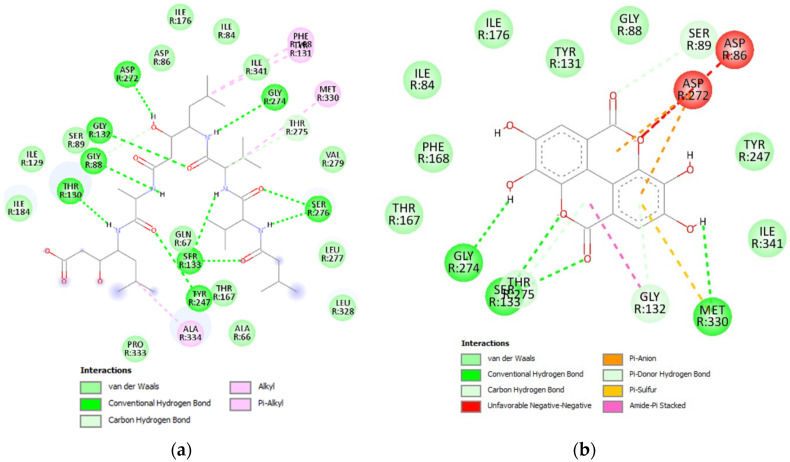
Two-dimensional diagram of interaction between RferPro27 protein and inhibitors, pepstatin and ellagic acid. (**a**) Interaction between RferPro27 and pepstatin A; (**b**) interaction between RferPro27 and ellagic acid.

**Table 1 insects-16-00421-t001:** The model quality assessment of the modeled digestive enzymes (DEs) and their corresponding reference structure.

Digestive Enzyme		pTM	ERRAT	Verify3D (%)	Ramachandran	MolProbity	QMean Disco Global	Root Mean Square Difference (Å)
Glycosidase	RferGly3	0.94	98.36	88.15	94.77% favorable	1.67	0.68	1.16
					0.16% outliers			
	6LGA (reference)	n/a	96.58	96.11	97.69% favorable	0.99	0.94	
					0% outliers			
Lipase	RferLip3	0.82	91.56	73.35	93.88% favorable	1.73	0.6	2.2
					0.81% outliers			
	7V55 (reference)	n/a	94.55	84.44	96.35% favorable	1.45	0.9	
					0% outliers			
Protease	RferPro27	0.85	88.5196	76.42	95.64% favorable	1.85	0.75	0.187
					0.27% outliers			
	5N7Q (reference)	n/a	90.7051	96.14	98.51% favorable	0.97	0.92	
					0.30% outliers			

## Data Availability

The datasets generated and/or analyzed during the current study are available in the NCBI repository under BioProject PRJNA275430 and SRA accession numbers SRR27695096, SRR27695095, SRR27695094, and SRR27695093.

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
