# Peer review of "Genome-Wide Identification and Expression Profiling of Glycosidases, Lipases, and Proteases from Invasive Asian Palm Weevil, Rhynchophorus ferrugineus"

_insects, 2025, doi:10.3390/insects16040421_

Round 1

Reviewer 1 Report (Previous Reviewer 2)

Comments and Suggestions for Authors

The authors did a great job revising the manuscript! It’s a completely different paper, and I feel that its approach is much clearer for me now. The figures also have much better quality now.

I noticed that in this version, the authors add a comparison with the genome assembly of the studied species, basically validating the predicted genes and their annotation. It is also a very nice addition to the study. I have a small question/suggestion to this analysis: is it possible to compare how many genes of each group are predicted in the genome and which proportion of those is expressed at each analyzed group? It might also be an interesting piece of information for further studies.

Comments on the Quality of English Language

I feel that the text could benefit from a proofreading round for the final version.

Here are some small text-related issues (definitely not an exhaustive list):

L40-41 “One such target is the digestive enzymes (DE), which impair plant feeding (herbivory).”: probably better “Digestive enzymes (DE), which impair feeding on plants (herbivory), are one of promising targets.”

L127 “RPWs”: please introduce the abbreviation “red palm weevil” at its first use.

L135 “anesthetized using CO2”: subscript missing from CO2

L204, L252, L981 “Lipase”: why is it capitalized?

L353 “three most highly expressing proteases” (and the same wording in several other places throughout the text): better “three proteases with the highest expression”

L370-374: I don’t fully understand why this sentence was deleted.

L1167 “proteas”: protease

Author Response

Reviewer 1- Point-by-point response.  

The authors did a great job revising the manuscript! It's a completely different paper, and I feel that its approach is much clearer for me now. The figures also have much better quality now.

Response#1 Thank you for the positive feedback.  

I noticed that in this version, the authors add a comparison with the genome assembly of the studied species, basically validating the predicted genes and their annotation. It is also a very nice addition to the study. I have a small question/suggestion to this analysis: is it possible to compare how many genes of each group are predicted in the genome and which proportion of those is expressed at each analyzed group? It might also be an interesting piece of information for further studies.

Response#2 We appreciate your suggestion. However, analyzing the proportion of each digestive enzyme group being expressed would be difficult because most of the genes in the R. ferrugineus are uncharacterized. Several additional steps of in silico validation would be necessary to identify how many glycosidases, lipases, and proteases are present in the weevil's genome, such as employing a gene prediction tool and domain searching tool against different databases specific to each enzyme type to validate the gene of having the necessary sequence and/or functional characteristics of a carbohydrate, lipid or protein digestive enzymes. However, we duly take note of this suggestion to be incorporated into further and deeper analysis of each digestive enzyme family within R. ferrugineus.

 Comments on the Quality of English Language

I feel that the text could benefit from a proofreading round for the final version.

Here are some small text-related issues (definitely not an exhaustive list):
L40-41 "One such target is the digestive enzymes (DE), which impair plant feeding (herbivory).": probably better "Digestive enzymes (DE), which impair feeding on plants (herbivory), are one of promising targets."

Response#3 Suggestion carried out. 

L127 "RPWs": please introduce the abbreviation "red palm weevil" at its first use.

Response#4 Suggestion carried out. 

L135 “anesthetized using CO2”: subscript missing from CO2

Response#5 Suggestion carried out. 

L204, L252, L981 "Lipase": why is it capitalized?

Response#6 Corrected in the revised manuscript (track changes marked, i.e., showing our edits).  

L353 "three most highly expressing proteases" (and the same wording in several other places throughout the text): better "three proteases with the highest expression"

Response# Suggestion incorporated.  

L370-374: I don't fully understand why this sentence was deleted.

Response#7 This sentence is included in the revised manuscript.  

L1167 "proteas": protease

Response#8 Corrected.

Reviewer 2 Report (New Reviewer)

Comments and Suggestions for Authors

The manuscript presents a comprehensive study on the identification and expression profiling of digestive enzymes in Rhynchophorus ferrugineus. The authors have employed a bioinformatics approach to analyze genomic and transcriptomic data, which is a timely and relevant topic given this pest's ecological and economic impact. The findings contribute valuable insights into the digestive mechanisms of this species and potential avenues for developing targeted pest control strategies. And I think it could be accepted after some minor revisions.

Introduction

The introduction provides a solid background on the significance of R. ferruginous as an invasive pest. However, it could benefit from a more detailed discussion of the implications of enzyme inhibition as a pest management strategy. Including specific examples of successful applications in other pests could strengthen the rationale for this study.

- Some words, like “in vitro” and “in vivo,” should be written in italics.

Materials and Methods

- Why was the sample from 2009 used? It has been 16 years since then. Will this affect the quality of the data generated?

- Lin162, is it too far to choose the genes of silkworms as the outgroup for building phylogenetic trees?

- Line 175, AMP?

Results

- Some figures are difficult to interpret due to low resolution or unclear labeling. Ensure that all visual data is presented clearly and at a high resolution.

Language and Formatting

- The manuscript is generally well-written, but there are a few grammatical errors and awkward phrasings that should be corrected for clarity. Thorough proofreading is recommended to improve overall readability.

Overall, this manuscript presents important findings that advance our understanding of the digestive enzymes in R. ferrugineus. With minor revisions addressing the points mentioned above, it has the potential to contribute to the field. I recommend that the authors carefully consider the feedback provided and revise the manuscript accordingly.

Author Response

Reviewer 2- Point-by-point response.  

The manuscript presents a comprehensive study on the identification and expression profiling of digestive enzymes in Rhynchophorus ferrugineus. The authors have employed a bioinformatics approach to analyze genomic and transcriptomic data, which is a timely and relevant topic given this pest's ecological and economic impact. The findings contribute valuable insights into the digestive mechanisms of this species and potential avenues for developing targeted pest control strategies. And I think it could be accepted after some minor revisions.

Response#1 Thank you for recommending our study for publication.

Introduction

The Introduction provides a solid background on the significance of R. ferruginous as an invasive pest. However, it could benefit from a more detailed discussion of the implications of enzyme inhibition as a pest management strategy. Including specific examples of successful applications in other pests could strengthen the rationale for this study.

Response#2 Thank you for the feedback. We have expanded the Introduction to reflect your suggestion better. The revised paragraph can be found in the Line 77 – 91 in the revised manuscript.

- Some words, like "in vitro" and "in vivo," should be written in italics.

Response#3 Suggestion carried out.

Materials and Methods

- Why was the sample from 2009 used? It has been 16 years since then. Will this affect the quality of the data generated?

Response#4 Rev#2 probably misunderstood the RPW insect collection and rearing section. We did not use the 2009 RPW samples for this. We mentioned here that the original red palm weevil collection was done in 2009 in the field, and thereafter, we maintained this culture in the rearing room. Lab-reared samples mentioned in this study originated from this pure-line culture. Field-collected samples were captured alive in May 2021 from the infested and removed date palm tree materials at Al Kharj in Saudi Arabia.

- Lin162, is it too far to choose the genes of silkworms as the outgroup for building phylogenetic trees?

Response#5 We chose silkworm (Bombyx mori) for several reasons. Firstly, it possesses a well-annotated genome. Moreover, from our observation of the BLAST hits, we found several BLAST hits annotated to silkworms but never from fruit flies (Drosophila melanogaster). Also, the silkworm, which is a lepidopteran, to be more closely related to coleopteran, than fruit fly which is a dipteran.  

- Line 175, AMP?

Response#6 We corrected this and other spelling errors or typographical mistakes.

Results

- Some figures are difficult to interpret due to low resolution or unclear labeling. Ensure that all visual data is presented clearly and at a high resolution.

Response#7 Higher resolution images 300 dpi have been included.

Language and Formatting

- The manuscript is generally well-written, but there are a few grammatical errors and awkward phrasings that should be corrected for clarity. Thorough proofreading is recommended to improve overall readability.

Overall, this manuscript presents important findings that advance our understanding of the digestive enzymes in R. ferrugineus. With minor revisions addressing the points mentioned above, it has the potential to contribute to the field. I recommend that the authors carefully consider the feedback provided and revise the manuscript accordingly.

Response#8 We thank the reviewer for his/her valuable suggestion; we have improved the quality of the English language and corrected language expression errors throughout the manuscript (track changes marked, i.e., showing our edit). The revised manuscript is now language editing done by a professional scientific language editing service by our university's Center for Scientific Writing in English. All changes made in the revised manuscript are highlighted with Track Changes marked (i.e., showing our edits).

Reviewer 3 Report (New Reviewer)

Comments and Suggestions for Authors

Overall, this is a very valuable paper presenting a detailed approach for the development of new insecticides by blocking important digestive enzymes in a (relatively) taxon specific manner. 

The writing is somewhat awkward in places, for example in the Abstract (lines 36-41) where the authors repeatedly state "We...".  I'd suggest varying the construct somewhat for readablility.

Line 122 "RNA later" should read "RNALater" (most likely an autocorrection error).

Line 127: "a single replicate" is an oxymoron. 

Lines 133 and 134 are incomplete or jumbled.

Line 140: "The HiSeq Illumina sequencing..." Does this refer to the instrument (e.g., HiSeq 2500) or to the sequencing chemistry?  Were paired end reads done? or Single reads?  What length?

Line 174.  I'm not sure what "correctly annotated" means.  Can this be clarified or simply deleted?

Comments on the Quality of English Language

See above comments.  The Abstract reads like it was machine-written, as do some sections of the Introduction.  Some rephrasing would help the manuscript enormously.

Author Response

Reviewer 3- Point-by-point response.  

Overall, this is a very valuable paper presenting a detailed approach for the development of new insecticides by blocking important digestive enzymes in a (relatively) taxon specific manner. 

The writing is somewhat awkward in places, for example in the Abstract (lines 36-41) where the authors repeatedly state "We...".  I'd suggest varying the construct somewhat for readablility.

Response#1 We have edited the abstract as the reviewer suggested.

Line 122 "RNA later" should read "RNALater" (most likely an autocorrection error).

Response#2 Suggestion incorporated.  

Line 127: "a single replicate" is an oxymoron. 

Ans# We have five samples of transcriptome data, and all annotated DEs mapped it onto the RPW genome and identified the candidate DEs.

Lines 133 and 134 are incomplete or jumbled.

Response#3 Sentence revised: "Insects were anesthetized using CO2 for 1–2 min, and the abdomen and gut were carefully dissected (male and female separately) under a light microscope".

Line 140: "The HiSeq Illumina sequencing..." Does this refer to the instrument (e.g., HiSeq 2500) or to the sequencing chemistry?  Were paired end reads done? or Single reads?  What length?

Response#4 Sequencing model details updated. Paired ends reads were done. Rev#3 probably missed Table S1. All information was included in Table S1. 

Line 174.  I'm not sure what "correctly annotated". Can this be clarified or simply deleted?

Response#5 We meant "manually annotated." The sentence was revised.

Comments on the Quality of English Language

See above comments.  The Abstract reads like it was machine-written, as do some sections of the Introduction.  Some rephrasing would help the manuscript enormously.

Response#6 As suggested by rev#3, we have now rephrased the manuscript (track changes marked).

This manuscript is a resubmission of an earlier submission. The following is a list of the peer review reports and author responses from that submission.

Round 1

Reviewer 1 Report

Comments and Suggestions for Authors

The overall aim of the study was to identify gut enzymes of Asian palm weevils which can be chemically targeted to control the weevil. To achieve this, the authors made four gut transcript libraries which they assembled. Following this, they identified the digestive enzymes in the assemblies and did a phylogenetic analysis. Finally a selected number of enzymes were bioinformatically analysed to determine ligand-binding specificities between the enzymes and potential inhibitors.

Overall, this study was not convincing. While the adult beetle feeds on the leaves of the plant, the larvae of the red palm weevil cause the most damage within the trunk of Date palm. In this case, why was the adult stage taken to review the plant degrading enzymes for insect pest management when it is more applicable for the larvae? In addition, there are numerous studies on the digestive enzymes of the red palm weevil available and thus it was not clear, why this specific study was conducted and how it increased knowledge of the system, as the focal enzymes were similar as those described in the previous studies.

This paper was not easy to follow due to numerous repetitions in the text and descriptions which could be much more clearly communicated in tabular form. In addition, the format of a merged results and discussion section increased the confusion.

There were numerous instances where the methods were extremely unclear. For example, there are a number of published genomes available for the weevil. Thus, why was a de novo assembly made instead of using the available sequence resource to quantify the transcripts and annotate them? If a metatranscriptomics approach was aimed at in this study, why were only enzymes of the insect gut identified? It did not seem as though an attempt was made to understand the production of plant degrading enzymes produced by the fungal community of the red palm weevil. Targeting fungal symbionts aiding in plant degradation could be another way of performing insect pest management.

The number of replicates for the differential expression analysis was not specified. As far as I could tell, a single sample was obtained from each treatment status. This sample number is very limited and thus not useful for any statistical comparisons. The samples were categorised into 4 groups, field collected male gut, field collected female abdomen, lab reared male gut and lab reared female abdomen. There was no reasoning as to why this was done, or why this kind of comparison was not followed through within the results/discussion. Furthermore, why was the female abdomen collected as opposed to female gut?

The choice of inhibitors for the enzyme-ligand interaction modelling was not substantiated well. For example ferulic acid is not a usual compound found in the wood. The analogue in lignin is coniferyl alcohol. Therefore the relevance of the inhibitors studied in silico is not clear.

Comments on the Quality of English Language

The English in this article is understandable, but in some instances gramatically incorect.

Reviewer 2 Report

Comments and Suggestions for Authors

The idea behind the submitted manuscript, to pave the way for developing an eco-friendly pesticide by studying the digestive enzymes of pests with transcriptomics, is very promising, and the approach is absolutely adequate for the goal. Therefore, I consider it a fair and valuable addition to the current body of knowledge. However, I have quite some concerns about data analysis and representation.

First, I am very skeptical about the differential expression analysis performed on groups of one sample per group (or I understood the description of this analysis incorrectly, which is also possible). Actually, I do not see this as a fundamental flaw of the analysis itself, as it was designed as exploratory, but rather suggest that the authors operate raw expression values without trying to perform differential expression on very limited data (or explain their idea behind the analysis more clearly if I understood it incorrectly).

Second, the illustrations within the main manuscript file are of very low quality and cannot be presented as such. Please update them, and I would be happy to assess them again.

Please find more particular questions and suggestions below.

The Materials and Methods section does not provide enough detail to be considered reproducible.

L100-101: “were collected from the Al-Kharj region of Saudi Arabia (24.1500° N, 47.3000° E) or maintained in our laboratory”: How were the animals collected in the nature? What was the origin of laboratory-maintained animals? How long were they maintained before extracting RNA?

L105 “Approximately 30 mg gut or abdominal tissues were collected”: how was the dissection procedure performed?

L117 “'hidden break' in the 28S RNA profile”: working with arthropod RNA, I understand what this means, but a link to a publication explaining this phenomenon to a wider audience would be very appropriate.

L134-136 “he criteria of significant differential expression were |log2 fold change |≥ 1(2), False Discovery Rate (FDR) ≤ 0.001 (3), and Bonferroni post hoc analysis.”: first, how was this analysis performed? I find it very at least unusual that one might retrieve any differentially expressed genes from comparing groups containing one sample. Second, I couldn’t understand what was meant by listing FDR and Bonferroni in the same sentence, as they are essentially different approaches to the same problem of multiple comparisons. I could imagine that FDR was used to correct for multiple comparisons of different genes, while Bonferroni correction could be additionally applied for comparisons of different groups of samples. Is it correct? Please consider adding a more detailed description of the procedures employed.

L141 “Heatmap package”: I’m not aware of the existence of a package with such a name. There is a heatmap() function in several packages, and there is a heatmap3 package (https://bmcbioinformatics.biomedcentral.com/articles/10.1186/1471-2105-15-S10-P16). Please make as clear as possible which package was used and provide a reference.

L153 “and their corresponding sequences were retrieved from NCBI”: please consider compiling a list of accession numbers for the supplement.

The Results section could also benefit from some clarifications.

L186-188 “Abdomen and gut transcriptome data (field and laboratory RPWs) were uploaded to

the NCBI SRA accession numbers:”: how many animals were there per group if only four runs were submitted to NCBI?

L201-202 “The BLASTx queries against the non-redundant database reported a collective of 22 glycosidases, 77 lipases, and 177 proteases”: how exactly was it determined that a particular protein belonged to a particular group (free text matching)? How many best matches were reported for each contig? I’m asking this question because in many cases the best match could be associated with a description like “putative protein XXX from species XXX”, while 2nd (or 3rd, or 5th) match contains a more useful description.

L203-205 “A higher number of differentially expressed genes (DEGs) were detected in the gut tissues than in the abdominal tissues.”: please provide a more detailed explanation if possible. Do you mean differentially expressed between males and females? Or field- and laboratory-reared? Why was the DEG analysis performed (what was its goal)? Where are the detailed results of the analysis?

Figures 1, 4 and 7: first, please consider highlighting the sequences obtained in this work for even easier understanding of the illustration. Second and more importantly, where could the reader actually access the sequences from R. ferrugineus? I see that the protein sequences are in Table S2, which is absolutely splendid, but if someone seeks to design primers for amplification of the coding sequence, they would rather need nucleotide sequences. I could find that the authors actually submitted the assembled transcriptome to the NCBI SRA database under the accession number GDKA01, which I could only commend. Would

Figure 1 is barely readable, while the other figures contain text that is just illegible. Please improve quality of figures makes them legible.

L262-263 “our previous study that profiled R. ferrugineus PCWDEs”: please spell out our PCWDE

Figs 3, 6 and 9 are just illegible and do not make any sense at their current quality.

Comments on the Quality of English Language

The manuscript contains numerous grammatical errors, which sometimes affect the readability. Most of these cases could have been prevented by using a simple (even automatic) proofreading tool. Here I provide some of the things I have noticed, but the list is definitely not comprehensive.

Title “The gut transcriptome profiling and in silico functional identification of glycosidases, lipases, and proteases from an invasive Asian palm weevil, Rhynchophorus ferrugineus”: “The” seems superfluous

L18-19 “discovering 22 carbohydrates, 77 lipids, and 177 protein-digesting enzymes”: “discovering 22 carbohydrate-, 77 lipid-, and 177 protein-digesting enzymes”?

L20-21 “the computer simulation of select enzymes”: “the computer simulation of selected enzymes”

L45-48 “Over four million species of the beetles (Order Coleoptera) known, representing spe-

cies-rich animals on earth is mainly attributed to the plant-feeding (herbivory) habit, as a

result of an extensive diversification of the plant cell wall degrading enzymes, besides the

functional specialization of key digestive enzymes [1,2].”

=>

“Over four million species of the beetles (Order Coleoptera) **are known at the moment**, **constituting the most** species-rich **animal group** on earth. **Their diversity???** is mainly attributed to the plant-feeding (herbivory) habit, as a result of an extensive diversification of the plant cell wall degrading enzymes, besides the functional specialization of key digestive enzymes [1,2].”

L57 “are herbivorous insect pest”: “are herbivorous insect pests”

L72-73 “These proteins may likely be targeted for disruption” => “These proteins may then / subsequently be targeted for disruption”?

L148-149 “A representative candidate of DEs was selected for each glycosidase, amylase, lipase,

and protease was chosen for comparative phylogenetic analysis” => “A representative candidate of DEs was selected for each group (glycosidases, amylases, lipases, and proteases) for comparative phylogenetic analysis”

L226 “This observation is to be expected”: “This observation is expected”?

L302-303 “this compound demonstrated insect mortality”: “this compound caused insect mortality” or “this compound was demonstrated to be pesticidal”

L342 “The pi-pi interactions is weaker”: “The pi-pi interactions are weaker”

L623 “S. Oryzae”: “S. oryzae”

Reviewer 3 Report

Comments and Suggestions for Authors

I have reviewed the manuscript “The gut transcriptome profiling and in silico functional identification of glycosidases, lipases, and proteases from an invasive Asian palm weevil, Rhynchophorus ferrugineus”. The study has identified digestive enzymes that are differentially expressed between the sexes of the species and has further validated some of these enzymes in silico. Overall, the study is interesting and well supported by the derived conclusions. However, the docking results need functional validation of at least one enzyme to strengthen the findings. 
